# Development and Validation of Cultural and Academic Experience Questionnaire: A study of East Asian Research Students at Australian Universities

**Yasir Latif \*, Neil Harrison and Hye-Eun Chu**

Department of Educational Studies, Macquarie University, Sydney 2109, Australia;
neil.harrison@mq.edu.au (N.H.); hye-eun.chu@mq.edu.au (H.-E.C.)
\* Correspondence: yasir.latif@mq.edu.au; Tel.: +61-447-670-786

**Abstract:** This article documents the development of a questionnaire designed to measure the cultural and academic experiences (CAEQ) of East Asian research students enrolled at Australian universities. The CAEQ comprises three subscales: sense of belonging, learning strategies and perception of progress. The scale was designed based on literature studies and ideas from previous scales. The target groups were doctoral students from Bangladesh, China, India, and Pakistan enrolled at Australian universities. Initially, 295 students responded to an online survey and 211 research students completed it. A factor analysis was conducted to explore the components of each scale. Confirmatory factor analysis confirmed that there are 2–4 significantly correlated components of each subscale. The developed scales in this study can be used by universities to monitor the academic progress and research progress of their international students from the East Asian region, along with ensuring that these students have improved cultural and academic experiences in the Australian higher education sector.

**Keywords:** East Asian students; scale-development; learning strategies; sense of belonging; perception of progress

## 1. Introduction

There has been a consistent decline in completion rates of international higher degree research (HDR) students at Australian universities since 2008. The Australian newspaper reported that "Australia's university dropout rate was worsening in all degree courses, with one in three students failing to complete their courses within six years of starting research course" [1]. Declining graduation rates are a direct threat to the 28 billion dollars that the Australian education sector earns annually as the third pillar of the Australian economy [2]. This economic imperative alone demonstrates a clear need to re-evaluate the international student experience at Australian universities.

The Australian Department of Education and Training (DET) focuses on various elements of student satisfaction, including physical facilities available on arrival in the country and the provision of living, learning and support facilities. However, relatively less consideration is given to other crucial issues, including the students' academic sense of belonging (SB), learning strategies (LS) and perception of progress (POP) in the Australian higher education context. SB, here, refers to the student's expectations that they will have a rightful and respected place in their host culture to integrate culturally and attain membership in semi-academic settings. LS refers to how students select, acquire, organize and integrate new knowledge into their own cognitive schemas. POP refers to the extent to which a student identifies the level of progress that is needed for graduation. Moreover, the definitions of the SB, LS, POP and their constructs are given below in Table 1.

**Table 1.** Definitions of variables and constructs of the cultural and academic experience questionnaire (CAEQ).

| Variables and Constructs | Definitions |
|---|---|
| Sense of belonging | The extent to which students feel personally accepted, respected, included, and supported by others in the school social environment [3]. |
| Cultural membership | An analysis of how social actors understand themselves as like or different from one another, and accompanying understandings of the social obligations entailed in these relationships of similarity and difference [4]. |
| Integration | The process of maintaining cultural heritage and participating in the wider society for successful adaptation [5]. |
| Learning strategies | The strategies used by students to select, acquire, organize, and integrate new knowledge into their own cognitive schemas. |
| Scientific learning | A learning approach that involves the attributes of thinking and doing [6]. |
| Artistic learning | A learning approach that involves the attributes of feeling and watching [6]. |
| Adaptive learning | A learning approach that involves the attributes of thinking and watching [6]. |
| Participative learning | A learning approach that involves the attributes of feeling and doing [6]. |
| Perception of progress | The way one perceives the level of academic and research progress required to successfully graduate. |
| Academic progress | Indicates the improvement in critical thinking, ideas development, dealing with academic issues and presenting excellent academic arguments. |
| Research progress | Indicates the progress in doing research, identifying research gaps, achieving major academic milestones, and learning techniques to conduct research. |

Cultural awareness, cultural sensitivity and differences in LS are key elements of student progress, but these elements are often overlooked in major government surveys of student satisfaction. The changes required to adapt to new learning strategies, along with developing a sense of belonging to the host culture, are not explored in these surveys. Loh and Teo [7] have suggested that culture has a strong impact on learning styles and Barron and Arcodia [8] have argued that international students must adopt the learning styles that are commonly applied in the host culture if they are to develop a SB to the host academic culture. SB plays a key role in the social and academic success of international students in Australia. For example, Levett-Jones and Lathlean [9] conducted a qualitative study about the academic progress of nursing students and found that those students who have a sense of belonging to their host culture and adopt appropriate learning strategies are more likely to graduate. The research team has explored the vast body of knowledge and conducted an in-depth systematic literature review of the previous scholarship, but remained unable to identify any scale which can serve the purpose. All of the literature review searches were completed through Google scholar, and later from other search engines such as Emerad, JStor, Science Direct, Taylor and Francis, Wiley Online Library, Blackwell, Ebscohost, Scopus, Elsevier, DOAJ, Copernicus, BASE, Core, Semantic Scholar. Many other search engines were also thoroughly explored. Moreover, Tables 2–4, presented later in this article, demonstrate the detail of the already developed measuring scales in literature. These scales measured the key variables of cultural and academic experience questionnaires including SB, LS and POP, but the research team remained unable to link these scales to the current study. This has led the research team to develop a new scale on the cultural and academic experiences of East Asian PhD students. Conclusively, the objective of this study is to develop a scale to measure the cultural and academic experiences of East Asian PhD students in Australian universities based on their sense of belonging, learning strategies and perception of progress".

## 2. Research Focus

The quantitative measurement of concepts requires social scientists to develop measurement scales based on the salient features, descriptions, specifications and properties of the concept. The cultural and academic experience questionnaire (CAEQ) is developed in this study to measure the experiences

of East Asian PhD students in Australian universities. The research team has explored, confirmed and validated the CAEQ after electronically administering it to East Asian research students. More specifically, this study addresses the research question, "what are the constructs constituting a sense of belonging, learning strategies and perception of progress for East Asian PhD students at Australian universities?"

## 3. Theoretical Framework

Over the past three decades, researchers like Gudykunst et al. [10] and Hofstede [11] have indicated that cultural differences promote uncertainty among international students in host countries, and, as a result, these students tend to avoid cultural interaction and integration. This avoidance impedes the chances of attaining a sense of belonging for these students with the host culture. SB focuses on mutually shared sociocultural attributes that include membership with the host group based on intercultural similarities and the integration of the differences in a mutually acceptable way. SB is important for East Asian research students because it promotes feelings of respect, adjustment, acceptance and comfort in interactions with the host community. SB also affects and frames the LS of students as suggested by Kolb [12] in his experiential learning theory, which suggests that intercultural differences affect learning styles.

Sense of belonging acts as an antidote to cultural differences which is supported by McMillan and Chavis's [13] indication that cultural membership and integration are the constructs of SB. It is convenient to presume that cultural membership and integration are likely to reduce cultural differences. In the current study, the face and content of SB items indicate the likelihood that the factor analysis of SB will generate cultural membership and integration as its constructs. In the case of learning strategies, Yamazaki [14] conducted a meta-analysis of the topography of learning styles to suggest that learning styles are relevant to the country of origin. Furthermore, Kolb and Kolb [6] have suggested that learning styles associated with a learning space can enhance academic performance, experiential learning, student development and faculty development and may influence learning strategies. This indicates that the adoption of learning strategies can facilitate better academic performance. Trigwell et al. [15] have explored student perceptions of their learning environment and found that students' approaches to learning, collectively contribute to their academic performance. Trigwell et al. [15] also found that Asian students focus largely on surface learning which leads them to poor learning and academic progress. The use of surface and deep learning varies according to differences in learning attitude, skills, goals, aspirations and learning styles [16]. Many researchers have tried to understand SB, LS and POP using quantitative methods. In this article, literature is critically analyzed to identify the possible utilization of previous studies in designing the CAEQ. A critical analysis of the previous scholarship is given below.

### 3.1. Sense of Belonging

The concept of sense of belonging is put forward by Maslow [17] in his hierarchy of needs theory. Goodenow [3] defined a sense of belonging as "the extent to which students feel personally accepted, respected, included and supported by others in the school social environment" (p. 80). Previous studies discussed above indicate that cultural integration, membership, social acceptance, academic motivation, academic adjustment, participation, engagement, interaction, support, empathy, affiliation and acceptance are important elements for a sense of belonging. Although many SB scales have been developed, when discussing the SB scales, the research team did not find these scales suitable for this study. An account of these scales has been given in Table 2, below.

**Table 2.** Scales used to assess sense of belonging.

| Scale | Purpose | Outcomes |
| --- | --- | --- |
| Goodenow [3] Psychological sense of school membership (PSSM) scale. | Perceived belonging; school membership in university students. | SB was strongly associated with class participation, university belongingness, social acceptance, and academic motivation for university students [18]. |
| Baker and Siryk [19] Student adaptation to college questionnaire (SACQ). | Improvement in the adaptation by detecting early signs of dropout based on personality, environment and controlling dropout using suitable remedial interventions. | SACQ identified students at risk by creating bonds between individuals and the institution. Ostrove and Long [20] found the SACQ to be a reliable tool. |
| Johnson et al. [21] National study of living-learning. Programs (NSLLP) survey on bachelor students. | Improvement in a social and academic environment for students through a sense of belonging. | NSLLP found that transitional social dimensions, perception of college racial climate, motivation, participation, and social integration had a strong relationship with SB. |
| Nichols [22] Psychological sense of school membership 2 scales (PSSM2). | SB impact on school engagement, dropout, participation, and interaction. | The results indicated no relationship between grades, social hierarchy, student-student and student-teacher relationships with a SB [22]. |
| Hoffman, Richmond, Morrow, and Salomone [23] scale. | Retention of students in college. | Concluded that faculty support, student-peer support, and interaction were constructs of SB [24]. |
| Ingram [25] College sophomore student survey. | Understanding the belonging of students to improve their outcomes. | Found that social belonging, academic belonging, and perceived institutional support were elements of a SB. College and social belonging were like each other. Diversity, peer-relatedness, and student support predicted belonging, whereas demographic variables did not [25]. |
| Leary and Baumeister [26] UCLA belongingness scale. | To see the relationship between school belongingness and achievement. | Found that school belongingness was related to school achievement and ongoing experience of interaction but unrelated to support [27]. |

As discussed earlier, the research team has remained unable to identify a scale on SB to incorporate the cultural and academic experiences needed for this study. It is noteworthy to mention that the psychological sense of school membership (PSSM) developed by Goodenow [3] has given the research team an idea to generate needed items for CAEQ by using literature along with expert opinions.

*3.2. Learning Strategies (LS)*

In this study, LS is defined as, how students select, acquire, organize and integrate new knowledge into their own cognitive schemas. Table 3 indicates the previous scales developed on LS.

**Table 3.** Previous scales used to measure the learning styles.

| Scale | Purpose | Outcomes |
|---|---|---|
| D. Kolb [28] Revised the reduced learning style inventory (RLSI), consisted of 9 items. | To measure and compare the relative preference of various learning styles in an individual. | RLSI was a reliable scale both in ranking and rating forms to measure individual learning style preferences Merrit and Marshall [29] but Fox [30] found RLSI unreliable. |
| Romero, Tepper, and Tetrault [31] Problem-solving style questionnaire (PSSQ), consisted of 14 items. | To address the shortcomings of the scale developed by Kolb [32]. | Improved test-retest reliability and internal consistency, convergent and discriminant validity, but Duff [33] could not find the internal consistency of the questionnaire. |
| Kolb [34] Learning style inventory (LSI), consisted of 12 items. | Used randomized items to control response biases. | Unsatisfactory results. |
| Kolb and Kolb [6] KLSI 3.1. version, consisted of 12 forced-choice items. | To improve the LSI inventory developed by Kolb [34]. | KLSI 3.1. was found to be a reliable and valid tool by Li and Armstrong [35]. |
| Manolis et al. [36] Reduced learning style inventory (RLSI), consisted of 17 items. | To verify the experiential learning model given by Kolb [34]. | RLSI was found to be fit, and three factors were loaded during factor analysis i.e., reflective observation-active experimentation (RO-AE), concrete experiences (CE), and abstract conceptualization (AC). |

Kolb and Kolb [6] introduced the experiential learning theory based on human learning methods which involve acts associated with watching, doing, feeling and thinking. Kolb and Kolb [6] have conceptualized these acts as reflective observation (RO), active experimentation (AE), concrete experience (CE) and abstract conceptualization (AC), respectively. Manolis et al. [36] developed a LS scale based on Kolb and Kolb [6] experiential learning theory, but there were two problems with this developed scale. First, the scale mixed RO and AE, which prevented vivid and appropriate quantification of the learning style. The second problem was associated with the psychometric capability of the measure. In our measure, the psychometric attributes of learning styles have been accounted for, without restricting ourselves to an interpretation of Kolb's learning styles—diverging, converging, accommodating and assimilating—as was the case for Manolis et al. [36].

*3.3. Perception of Progress (POP)*

Progress means working towards an objective or goal [37]. In this study, POP is interpreted as working towards the graduation goal. Perception of progress (POP) is presented in the literature together with many similar terms such as academic performance [38], performance expectancy [39], achievement goal [40], goal investment [41], persistence and perceived accomplishment [42]. The current study defines POP as the extent to which a student perceives the progress required to successfully graduate. This study expresses POP in terms of academic progress and research progress. According to Bryan and Locke's [43] suggestion that knowledge of goal is a predictor of improved performance, POP can only help research students to effectively navigate toward their graduation when the research students are conscious of their graduation goal. We have devised a scale to measure students' POP based on their SB and LS, since SB and LS are strong predictors of a student's retention and progress [44]. These are defined in Table 4.

**Table 4.** Scales on the perception of progress.

| Scale | Purpose | Outcomes |
|---|---|---|
| S. D. Brown, Lent, and Larkin [45] Scale on educational requirements and academic milestones. | To understand the impact of knowledge of educational requirements and milestones on students' performance. | The scope of these instruments was limited to mathematics, science and engineering career prospective. |
| Wolters [46]; Chapell et al. [47] Behavioral self-regulation scale. | Using cumulative grade point average CGPA for behavioral self-regulation for improved performance. | Developed a highly reliable performance improvement scale based on self-regulation. |
| Chemers, Hu, and Garcia [48] Academic self-rating scale, academic evaluation scale, and academic expectation scale. | Helps individuals to understand their self-performance, academic performance in university and to measure expected performance. | Concluded that academic self-efficacy and performance were unrelated to performance and adjustment. |
| Elliot and McGregor [49] Achievement goals questionnaire (12 items). | Focuses on school-level performance, performance approach, performance-avoidance, mastery approach, and mastery avoidance. | Concluded that this scale is reliable and valid to measure students' school performance Pekrun, Elliot, andMaier [50]. |
| Chauvin, Demont, and Rohmer [51] School social judgment scale (SSJS). | To develop a self-report measure of social justice for children at school to address 12 behaviors. | Concluded that the scale was reliable. It only rated self-performance of students in the social setting of school and did not include academic and research settings. |

An analysis of the previous scholarship indicates that there is a need for a valid and reliable instrument to measure the cultural and academic experiences of East Asian research students at Australian universities. Three hypotheses can be drawn from the above debate:

**Hypothesis 1.** *Cultural membership and integration are the valid and reliable constructs of a SB.*

**Hypothesis 2.** *Scientific, artistic, adaptive and participative learning are the valid and reliable constructs of LS.*

**Hypothesis 3.** *Academic progress and research progress are the valid and reliable constructs of POP.*

## 4. Methodology

### 4.1. Sample

The population of East Asian research students is continuously changing in Australian universities. It is not possible to estimate the exact number of East Asian research students studying at Australian universities. These circumstances have forced the research team to estimate the sample size using item response theory (IRT). The sample size was calculated based on item response theory, in which participant numbers must be five times the variables as per Bentler and Chou [52], and Hair et al. [53]. Initially, there were 41 research items at the time of the survey, and the sample size calculated was 205, that is $41 \times 5 = 205$, but the collected sample was 211 which was enough to further analyze the scale. Of the 211 respondents, 62% were male, 36% were female and 2% did not indicate their gender, but no participant identified as transgender. Participants were from Bangladesh (24), China (51), India (20) and Pakistan (116). About 60% of students were 21–30 years of age, 35% were between 31 and 40 years and 6% were above the age of 40. These participants were studying in various states across Australia, including New South Wales (106), Queensland (21), Victoria (31), Western Australia (8), Tasmania (13), South Australia (2), Queensland (21) and the Australian Capital Territory (9). Some students had obtained Australian residency (20%), but most (80%) were student visa holders. Before

enrolling in their Australian courses, 80% of the students had studied in an urban higher education institution in their home country, 18% had completed their studies in rural locations and 2% studied at remote locations in their home country. These participants spoke various languages, including Bengali (24), English (126), Hindi (1), Mandarin (34), Pashto (1) and Urdu (25). Over half of the students (69%) focused only on their research studies, whereas about 30% were working as tutors and research assistants in universities, with about 1% in administrative roles. Most (93.7%) were studying full time and 6.3% were part-time students. These students were enrolled in various academic and research disciplines, including engineering (36%), business (20%), science and medicine (21%) and social sciences (13%) and the remainder (10%) in other disciplines. Most (90%) were in PhD programs, and only 10% were in masters leading to PhD programs. Among the participants, 51.2% were married, 47.9% were unmarried and 0.9% were divorced.

### 4.2. Instrument Development and Research Design

#### 4.2.1. Content and Face Validation

After doing an extensive literature review of the items in the previous measures, the research team got a clear understanding of the needed items for the cultural and academic experience questionnaire (CAEQ). All the items demonstrated the definition and the deep meaning associated with their relevant concepts. First, all these items were created on the basis of the literature review, followed by further improvement through five in-depth brainstorming sessions. Consequently, 41 items were identified. These items focused on three main variables including SB (15 items), LS (16 items) and POP (10 items). After item formation, five experts on international students' experiences in Western countries were identified within the university research team, and the items were sent to these experts for their comments related to the face and content validity of the items. These expert professors provided positive comments on the suitability of each item for the SB, LS and POP scales. The scales were emailed to five other international experts. All of the five experts gave their approval in verbal and written form, related to the suitability of each item of the scale. After a two-fold expert professor's approval process, the research team invited a group of five PhD students via an advertisement on the university notice board for a group discussion related to the suitability of the scale. Three Pakistani and two Chinese students offered to participate. These five students indicated that they did not find any difficulties in understanding the items and indicated that the scales covered most of their cultural and academic experience in Australia. All the experts and PhD scholars approved the scale for the survey data collection.

#### 4.2.2. Process of Scale Development

To develop and validate the items, we followed the process given in Figure 1. The definitions of variables including SB, LS and POP were provided to the participants at the beginning of each scale to help them answer the online questionnaire in the same context as intended by the research team. These items were theoretically hypothesized to measure constructs of SB, LS and POP. After the data collection, the exploratory and confirmatory factor analysis were separately conducted on SB, LS and POP scales to identity their respective constructs. Initially, 15 items for SB, 16 items of LS and 10 items for POP were developed. Four response options comprising strongly disagree (SD = 1), disagree (D = 2), agree (A = 3) and strongly agree (SA = 4) were provided. The neutral option was purposefully avoided to maximize the efficiency of the survey on a very limited population of East Asian PhD students.

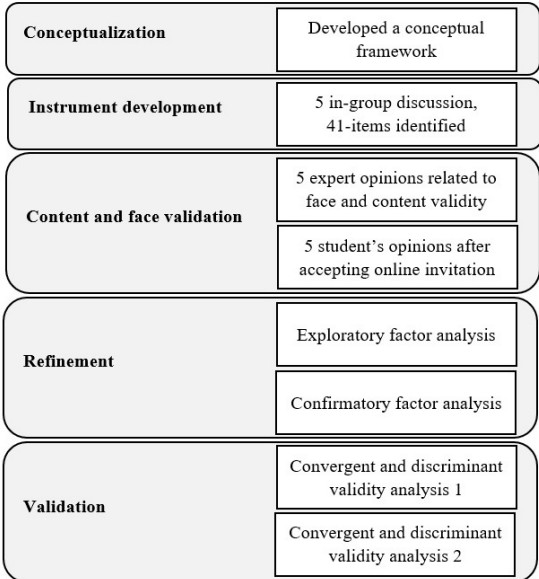

**Figure 1.** Scale development process.

### 4.3. Data Collection

Ethics approval was obtained from the university ethics committee of the research team. An online Qualtrics survey was advertised on Facebook and LinkedIn groups of East Asian PhD students in Australia. Snowball sampling was used to connect with more participants following the recommendations and suggestions of those participants who already participated in the survey. Feedback from the participants played a key role in accessing the unknown and continuously varying population size of East Asian research students at Australian universities. The research team found snowball sampling as the most suitable technique unknown and widely scattered population of East Asian research students throughout Australia. The research team, therefore, advertised the research project on Facebook and LinkedIn groups related to East Asian students in Australian universities for their voluntary participation. For the survey advertisement, only people who had already spent 6 months in their PhD candidature were requested to participate. The mandatory time limit of six months was set to ensure that the students had attained sufficient cultural and academic experience in Australian universities. Luckily, most of the survey participants referred their colleagues for participation in the survey. A total of 297 research students responded to the invitation to take part in this component of the research, but 86 did not complete the survey, leaving a total of 211 respondents in the final data set. Within these data, two responses given by participants were missing and were completed using multiple imputations because we regarded this as the most appropriate method for dealing with missing responses [54].

### 4.4. Data Analysis

Exploratory factor analysis (EFA) and confirmatory factor analysis (CFA) were used for item exploration and confirmation of item structure in the CAEQ. SPSS 24 was used for EFA, whereas AMOS 24 was used for CFA. EFA was used to identify meaningful clusters of items that make constructs of each of the SB, LS and POP [55] to identify the meaningful clusters of the items [56]. EFA is directly relevant for evaluating a scale's internal structure and helped in the identification of the factors in CAEQ. CFA confirms and validates EFA if the same clusters of the items are completely or partially loaded in CFA results. In this way, EFA and CFA also provide information about the internal consistency of the scale and thus, contributed to the analysis of CAEQ. The output of CFA was used for checking convergent and discriminant validity for the final verification of the clusters of items. Two second-order tests guided by Chen and Paulraj [57] and Van Dyne and LePine [58] were conducted

after the convergent and discriminant validation check. These tests used an approach in which the standard errors (SE) must be less than the unstandardized regression weights for every item in order to prove their convergent validities according to Gerbing and Anderson [59]. The results are given in Table A1 in the Appendix A. For this test, the stats tool package was used, which was developed by Gaskin [60] to check convergent and discriminant validity.

## 5. Findings

### 5.1. Exploratory Factor Analyses (EFA)

#### 5.1.1. Sense of Belonging (SB)

EFA was conducted on the 15 items for SB using varimax rotation under principal component analysis (PCA) and Kaiser Meyer Olkin (KMO) was in an acceptable range with a value of 0.70 [61], Bartlett's test of sphericity was satisfactory ($p < 0.001$) for two factors i.e., cultural membership and integration were exceeding unity with eigenvalues of 2.49 and 1.40, respectively, accounting for 55.6% of the variance for SB, that is cultural membership and cultural integration. Cultural membership and cultural integration were defined earlier in this article in Table 1. All items in both factors loaded above 0.50. Factor 1 was membership, which accounted for 35.53% of the variance and had four items (SB1, SB2, SB5 and SB15) related to cultural membership. Factor 2 was integration, which accounted for 20.03% of the variance and had three items (SB11, SB12 and SB13) related to integration. Several items (SB3, SB4, SB6, SB7, SB8, SB9, SB10 and SB14) were removed during EFA because these items were loading on multiple constructs, as well as giving less factor loading value in the communalities table in EFA output. The overall Cronbach's alpha reliability of SB was 0.70. The cut-off values for factor analysis are given in Table 5.

**Table 5.** Exploratory factor analysis: Sense of belonging. Extraction method: Principal component analysis. Rotation method: Varimax with Kaiser normalization. Cronbach's alpha > 0.5 is moderate and acceptable [2] (p. 363). Alpha = 0.70 is recommended by Nunnally and Bernstein [62].

|  | Factors | Eigenvalue | Items | Factor Loadings |
|---|---|---|---|---|
| **SB**<br>**($\alpha = 0.70$)** | Cultural membership<br>($\alpha = 0.67$) | 2.49 | SB2. I feel that I am accepted by students from other cultural groups | 0.76 |
|  |  |  | SB1. I socialize with students from other cultures. | 0.81 |
|  |  |  | SB15. I feel that my research colleagues respect my academic views. | 0.57 |
|  |  |  | SB5. I feel capable of speaking to and interacting with lecturers and professors at my university. | 0.52 |
|  | Cultural integration<br>($\alpha = 0.63$) | 1.40 | SB13. My behavior and language are becoming accustomed to Australian culture. | 0.79 |
|  |  |  | SB12. I am adjusting my behavior and language to Australian culture. | 0.75 |
|  |  |  | SB11. In Australia, I feel more comfortable doing what I want. | 0.65 |

#### 5.1.2. Learning Strategies (LS)

EFA was conducted on 16 items for LS using varimax rotation under principal component analysis (PCA). KMO was in an acceptable range with a value of 0.74 which is above 0.60 [62,63]. Bartlett's test of sphericity was satisfactory ($p < 0.001$) for four factors i.e., scientific learning, artistic, adaptive and participative learning strategies, and exceeded unity with eigenvalues of 3.23, 2.06, 1.54 and 1.23, respectively, accounting for 53.68% of the variance for LS which are scientific learning, artistic learning, adaptive learning and participative learning. These factors are presented in Table 6 and defined above in Table 1. Factor 1 was scientific learning and accounted for 21.55% of the variance, with five items



(LS1, LS12, LS13, LS14 and LS16). Factor 2, artistic learning style, accounted for 13.70% of the variance with four items (LS5, LS6, LS7 and LS11). Factor 3 was an adaptive learning style and accounted for 10.24% of the variance, with three items (LS2, LS3 and LS4). Factor 4 was a participatory learning style that accounted for 8.18% of the variance and had three items (LS8, LS9(R) and LS10). Here (R) in LS9(R) indicates the reverse coded item. All items had loadings above 0.6, other than one item, LS14, which had loaded above 0.4. The only item removed during EFA was LS 15. The overall coefficient alpha for LS was 0.70.

**Table 6.** Exploratory factor analysis: Sense of belonging. Extraction Method: Principal component analysis. Rotation method: Varimax with Kaiser normalization. Reliability "$\alpha$" > 0.5 is moderate and acceptable Hinton et al. [64] (p. 363). Reliability "$\alpha$" = 0.70 is recommended by Nunnally and Bernstein [62].

| | Factors | Eigenvalue | Items | Factors Loadings |
|---|---|---|---|---|
| **LS** ($\alpha$ = **0.70**) | Scientific learning ($\alpha$ = 0.691) | 3.23 | LS12. I find myself questioning things I hear or read, relevant to my research study, to decide if I find them convincing. | 0.78 |
| | | | LS13. When a theory, interpretation, or conclusion is presented in a research meeting or in the literature, I try to decide if there is good supporting evidence. | 0.77 |
| | | | LS1. I prefer to pose my own question and work out my own answers. | 0.65 |
| | | | LS16. Whenever I read or hear an assertion or conclusion relevant to or closely related to my research study, I think about possible alternatives. | 0.62 |
| | | | LS14. I treat the literature review as a starting point and try to develop my own ideas about it. | 0.42 |
| | Artistic learning ($\alpha$ = 0.66) | 2.06 | LS6. I expect that my supervisors guide me in every step of my research process. | 0.73 |
| | | | LS11. I wait until my supervisors tell me how to solve a problem to save me time. | 0.70 |
| | | | LS5. I follow the footsteps of academic authority in my domain e.g., supervisor. | 0.65 |
| | | | LS7. I expect that my supervisors choose my research topic. | 0.62 |
| | Adaptive learning ($\alpha$ = 0.58) | 1.54 | LS3. I learn through trial and error to solve my research problems. | 0.75 |
| | | | LS4. I learn through repetition. | 0.67 |
| | | | LS2. I learn when I work (in the discussion, collaboration) with other students. | 0.66 |
| | Participative learning ($\alpha$ = 0.53) | 1.23 | LS8. I always ask questions when I am unsure. | 0.71 |
| | | | LS9(R). I feel embarrassed to ask for help in front of my research colleagues. | 0.71 |
| | | | LS10. My supervisor's approach to learning suits me. | 0.63 |

### 5.1.3. Perception of Progress (POP)

EFA was conducted on 10 items for POP using varimax rotation under principal component analysis (PCA), KMO was in an acceptable range with a value of 0.89 [63], Bartlett's test of sphericity was satisfactory ($p < 0.001$) for two factors of perception of progress i.e., academic progress and research progress exceeded unity with eigenvalues of 5.24 and 1.13, respectively, accounting for 64.20% of the variance for POP. The first Factor 1 is academic progress and Factor 2 is research progress. These factors are already defined in Table 1. Table 7 indicates that all items for both factors loaded above 0.60. Factor 1 was academic progress which accounted for 52.93% of the variance and had six academic development items (POP5, POP6, POP7, POP8, POP9 and POP10). Factor 2 was research development

which accounted for 11.26% of the variance and had three items (POP1, POP2 and POP3). The overall coefficient alpha for POP was 0.89.

**Table 7.** Exploratory factor analysis: Perception of progress. Extraction method: Principal component analysis. Rotation method: Varimax with Kaiser normalization. Reliability "α" > 0.5 is moderate and acceptable Hinton et al. [64] (p. 363). Reliability "α" = 0.70 is recommended by Nunnally and Bernstein [62].

|  | Factors | Eigenvalue | Items | Factor Loadings |
|---|---|---|---|---|
| **POP** **(α = 0.89)** | Academic progress (α = 0.87) | 5.24 | POP5. I think I am confident when I discuss my research with my supervisors. | 0.80 |
|  |  |  | POP8. My critical thinking has improved throughout my studies. | 0.77 |
|  |  |  | POP9. I can now deliver sound arguments at research events. | 0.77 |
|  |  |  | POP6. I think I am confident when I discuss my research with my fellow students. | 0.72 |
|  |  |  | POP7. Compared to the time when I first started my research study, I am now better at dealing with issues relating to research. | 0.71 |
|  |  |  | POP10. I have a clear idea of how I need to analyze my data. | 0.68 |
|  | Research Progress (α = 0.81) | 1.13 | POP1. I think I have identified the research gap in my area of research. | 0.88 |
|  |  |  | POP2. I think I know how to address the gap in my research. | 0.87 |
|  |  |  | POP3. I have been achieving major milestones for my degree. | 0.61 |

### 5.2. Confirmatory Factor Analysis (CFA)

Preliminary exploratory factor analysis and confirmatory factors revealed that measures used in this research displayed adequate psychometric properties. The model fit of the CFA for the research conducted was acceptable, with $\chi^2/df = 662/417 = 1.59$, ($p < 0.000$); comparative fit index (CFI) = 0.88; root mean square error of approximation (RMSEA) = 0.05; standard root mean square residual (SRMR) = 0.074; PCLOSE = 0.28, GFI 0.83, AGFI = 0.80, IFI = 0.881 TLI = 0.86. Most of the factor loadings were significant and surpassed the 0.7 level [62] and the more conservative cut-off value of 0.5 or higher, the value recommended by Hair Jr et al. [65]. In contrast, Ertz et al. [66] recommended 0.4 as the cut-off value for factor loadings. The model for indices suggested that the analysis supported a good fit between the eight factors model and the data, which are scientific learning, artistic learning, adaptive learning, participative learning, cultural membership, integration, academic progress and research progress. The findings will be reported in the next section. Following the CFA methodology adopted by Varshneya and Das [67], the newly developed scales were reliable because the coefficient alpha and composite reliability (CR) values were in the acceptable range of 0.6–1.0 [62,68,69]. The alpha and CR of variables were in an acceptable range. The alpha for both LS and SB was 0.70, and 0.90 for POP. CR was 0.70 for LS and SB, whereas it was 0.91 for POP. SB had an alpha equal to 0.68 and CR equal to 0.67. POP had an alpha value equal to 0.89, and CR equal to 0.91. The CFA pathway diagram along with standardized coefficients is given in Figure 2.

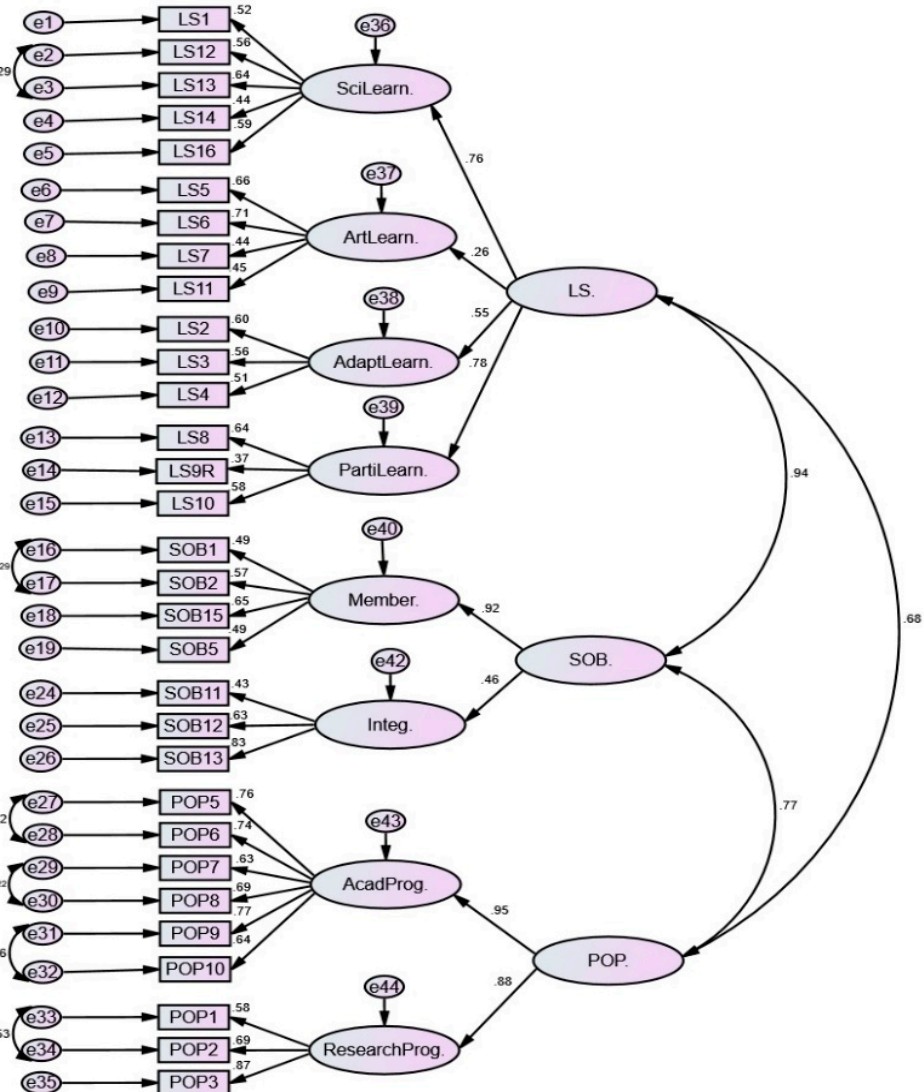

**Figure 2.** Confirmatory factor analysis structure. The values in Figure 2 are standardized coefficients.

*5.3. Convergent and Discriminant Validation*

Brown [70] stated that convergent validity indicates strong correlation items within a factor, whereas discriminant validity indicates that factor loadings are not highly correlated within factors, indicating that hypothesized items are different from each other. In this way, convergent and discriminant validity ensures that items that are different from one another, still focus on the same variable.

5.3.1. Convergent Validity

For convergent validity, there were two conditions. The first was that the standardized loadings of the items given in Figure 2 should be above 0.4, but in this study, the loading of LS9 (R) was less than 0.4 (0.37). Despite this, the item was kept due to its importance for the measure. The item measures students' feelings about asking questions in front of their research colleagues. The second condition was that the average variance extracted values (AVE in Table 8) should be above 0.5, but in this study, AVE for LS was less than 0.5 (0.39), suggesting that convergent validity was not satisfied according to the criterion of Gerbing and Anderson [59].

A second-order test was used to assess convergent validity following the method adopted by Chen and Paulraj [57]. In this test, the twofold standardized error (SE) of unstandardized regression

estimates must be less than the standardized regression estimates. For example, 2(SE) < regression estimates, for example 0.528 < 1.187 for member in Table A1. The twofold SE was the lesser, so the convergent validity was established [71], as shown in Table A1 in the Appendix A. All item loadings indicated in Figure 2 were significant ($p < 0.01$) and the values of average variance extracted from sense of belonging and perception of progress were above 0.50 (see Figure 2) in support of convergent validity [60].

**Table 8.** Analysis values on convergent and discriminant validity (N = 211).

| | Values for Convergent Validity | | | | Values for Discriminant Validity | | |
|---|---|---|---|---|---|---|---|
| | CR | AVE | MSV | MaxR(H) | LS | SB | POP |
| **LS** | 0.69 | (0.386) * | (0.885) ** | 0.771 | (0.621) *** | | |
| **SB** | 0.67 | 0.531 | (0.885) ** | 0.856 | 0.941 | (0.729) *** | |
| **POP** | 0.91 | 0.839 | 0.594 | 0.927 | 0.682 | 0.771 | 0.916 |

* Average variance extracted (AVE) < 0.5 indicating convergent validity issue. ** Maximum shared variance (MSV) < AVE indicating convergent validity issue. *** Lower self-correlations of variables than the correlation with other variables indicating discriminant validity issue, for example (LS←→LS < LS←→SB). Note: The values in Table 8 are generated by putting the correlation estimates along with their regression weights generated by AMOS in the stats tool package developed by Gaskin [60].

### 5.3.2. Discriminant Validation

The average variance extracted for learning strategies was slightly below the threshold, that is, AVE = 0.386 (see Table 8). The values of the square root of the AVE for learning strategies ($\sqrt{0.386} = 0.621$ < 0.941 and $\sqrt{0.386} = 0.621 < 0.682$) and sense of belonging ($\sqrt{0.531} = 0.729 < 0.941$ and $\sqrt{0.531} = 0.729 < 0.771$) were lower than the value of the correlations with another factor suggesting that the discriminant validity between the factors was not established. The discriminant validity was initially proved by the chi-square difference ($\Delta\chi2$). Results showed that there was significant chi-square difference ($\Delta\chi2$) between learning strategies and sense of belonging ($\Delta\chi2 = 347.1/200-516.3/201 = 0.83$, df = 1; $p < 0.05$). There was significant chi-square difference ($\Delta\chi2$) between SB and POP ($\Delta\chi2 = 143.15/94 - 296.9/95 = 1.6$, df = 1; $p < 0.05$). Similarly, there was significant chi-square difference ($\Delta\chi2$) between LS and POP ($\Delta\chi2 = 404.5/240 - 542.14/241 = 0.565$, df = 1; $p < 0.05$). Table A2 in the Appendix A presents the chi-square difference table.

Based on the reliability statistics given in Tables 2–4 for the constructs of SB, LS and POP, all of the constructs are reliable. The results of convergent and discriminant validity, presented in Tables A1 and A2, indicate that all the constructs of SB, LS and POP hold their respective items converging on their constructs to ensure convergent validity. Similarly, all items in each construct of SB, LS and POP are different from each other to ensure discriminant validity. Hence, H1, H2 and H3 are proven.

## 6. Discussion

The purpose of this study has been to develop and validate a new scale (CAEQ) that measures the cultural and academic experiences of East Asian research students. Two factors of sense of belonging, including membership and integration, were previously suggested by Goodenow [3] (see Table 5), as well as four factors related to learning strategies (see Table 6) and two factors related to the perception of progress (see Table 7). All clusters were loaded on their relevant constructs. Items in each cluster were loaded on their relevant constructs, and were not loaded across other clusters. In terms of the scale reliability, examination of coefficient alpha (see Tables 5–7) and CR value (see Table 8) showed that all the scales and sub-scales of sense of belonging, learning strategies and perception of progress were reliable. The reliability of adaptive learning and participative learning was relatively lower (see Table 6), but the values were still within an acceptable range. All the items in each construct of each variable were statistically different from each other and designed to converge on their relevant constructs in order to establish convergent and discriminant validity, see Tables A1 and A2 in the

Appendix A. All scales were found to be reliable, valid and coherent with the definitions of their constructs and variables.

Goodenow's [3] theoretical predictions and statistical verification methodologies adopted by Chen and Paulraj [57]; Van Dyne and LePine [58]; Gerbing and Anderson [59] and Gaskin [60] were used to identify that all the variables including SB, LS and POP were statistically different from each other. The constructs of SB, LS and POP were different from each other and were converging on their relative variables i.e., cultural membership and integration were different from each other and converged on SB. The items of cultural membership focused on cultural acceptance, socialization, respect and interaction. The items of integration focused on adjustment, feeling comfortable and behavioral and linguistic modification. The interpretation of two factors of SB (cultural membership and integration) was coherent with the definitions in the literature [4,5]. We found that attaining cultural membership in an Australian university context is relatively more important to attaining POP than feeling a sense of integration. This is because cultural membership promotes a sense of belonging for East Asian research students within the Australian academic system, whereas cultural integration functions to increase the process of adaptation with the host culture while maintaining connectivity with the home culture. Hence, integration was not identified as relatively less important than cultural membership to graduate successfully in Australian higher education. However, integration helps international research students to connect with the host academic culture without completely distancing themselves from their home academic culture [72].

We also found that four factors of LS were psychometrically coherent with the four quadrants of Kolb's learning cycle [12] and were consistent with the definitions of those quadrants (see Table 1). Similar to previous research, the first factor, scientific LS, was linked with inferring, reasoning, experimental reliance, understanding reasoning and interlinking various concepts. Artistic learning, as the second factor, was based on feeling attributes of human nature, leading East Asian research students to feel dependent on role models for learning by listening and observing. The third factor was an adaptive LS based on attributes of thinking and watching so its items were focused on learning through repetition, watching others and using trial and error methods. The fourth factor was participatory learning and was based on attributes of feeling and doing. These items focused on discussions to remove uncertainties, asking for help and sharing knowledge for learning. Finally, the two factors of POP were coherent with the definition. The first factor was academic progress and its items focused on gaining confidence, dealing with research issues, improving analytical abilities, delivering academic arguments and improvements in critical thinking. The second factor was research progress and its item focused on achieving research milestones by identifying and addressing research gaps [73]. All the items and variables gave results, as indicated in the literature, to establish the validity of the CAEQ.

Theories on self-efficacy, self-regulation and goal achievement of students in international destinations predict that perception of progress corresponds to a sense of belonging with the host culture and adopting to learning strategies (see standardized correlation coefficient in Figure 2). This study validates the relationship between the variables of interest as these relationships are in line with predictions of previous theories. These findings provide a construction of validity for the CAEQ, which implies that the POP of East Asian research students in Australian universities depends on their SB to the new academic culture and adopting learning strategies. It is reasonable to assume that cultural and academic experiences play a key role in the POP of East Asian research students at Australian universities. This implies that universities should develop an academic system in such a way that it becomes more open and responsive to East Asian research students in order for them to feel a stronger SB to the Australian academic culture and, in turn, start interacting with their research colleagues to learn those learning strategies that will improve their chances of success.

## 7. Conclusions

It is concluded that cultural membership and integration are reliable and valid constructs for sense of belonging. Scientific, artistic, adaptive and participative learning are reliable and valid constructs of

the learning strategies. It is also concluded that academic progress and research progress are important, valid and reliable constructs of POP. The relationship between SB, LS and POP is yet to be determined. Future research can be conducted on the impact of SB and LS on the POP of international research students in national and international contexts. It is also recommended to use the CAEQ across the globe to re-verify its reliability and validity.

## 8. Policy Implications

It is essential for universities and policy-making bodies in Australia to consider the importance of academic and social bonding of East Asian research students studying at Australian universities. East Asian research students must be given an opportunity, as a part of their research candidature, to understand, interact and connect with people and systems that are related to their area of research. These opportunities can give them academic leverage in conducting their research more precisely in line with current demands, issues and trends of Australian higher education scholarships.

**Author Contributions:** This research is conceptualized by Y.L. whereas N.H. and H.-E.C. has provided detailed help in conceptualizing, reviewing and guiding as primary and adjunct PhD supervisors respectively. All authors have read and agreed to the published version of the manuscript.

**Funding:** This research received no external funding and the APC was paid by Y.L.

**Conflicts of Interest:** The authors declare no conflict of interest.

## Appendix A

**Table A1.** Convergent validity using unstandardized regression weights.

|  |  | Estimate | Standard Error (S.E.) | 2 (S.E.) | Estimates > 2(S.E.) |
|---|---|---|---|---|---|
| PartiLearn | LS | 1 |  | 0 | Accepted |
| Member | SB | 1.187 | 0.26 | 0.528 | Accepted |
| AcadProg | POP | 0.793 | 0.11 | 0.218 | Accepted |
| ResearchProg | POP | 1 |  | 0 | Accepted |
| ArtLearn | LS | 0.276 | 0.12 | 0.238 | Accepted |
| AdaptLearn | LS | 0.629 | 0.18 | 0.352 | Accepted |
| SciLearn | LS | 0.89 | 0.2 | 0.394 | Accepted |
| Integ | SB | 1 |  | 0 | Accepted |
| LS16 | SciLearn | 1 |  | 0 | Accepted |
| LS14 | SciLearn | 0.709 | 0.15 | 0.294 | Accepted |
| LS13 | SciLearn | 0.919 | 0.15 | 0.306 | Accepted |
| LS12 | SciLearn | 0.837 | 0.15 | 0.302 | Accepted |
| LS1 | SciLearn | 1.003 | 0.18 | 0.366 | Accepted |
| LS11 | ArtLearn | 1 |  | 0 | Accepted |
| LS7 | ArtLearn | 1.038 | 0.25 | 0.5 | Accepted |
| LS6 | ArtLearn | 1.707 | 0.35 | 0.698 | Accepted |
| LS5 | ArtLearn | 1.442 | 0.29 | 0.588 | Accepted |
| LS4 | AdaptLearn | 1 |  | 0 | Accepted |
| LS3 | AdaptLearn | 1.018 | 0.23 | 0.462 | Accepted |
| LS2 | AdaptLearn | 1.229 | 0.28 | 0.552 | Accepted |
| LS10 | PartiLearn | 1 |  | 0 | Accepted |
| LS8 | PartiLearn | 1.163 | 0.21 | 0.428 | Accepted |
| SB15 | Member | 1 |  | 0 | Accepted |
| POP10 | AcadProg | 1 |  | 0 | Accepted |
| POP9 | AcadProg | 1.127 | 0.11 | 0.214 | Accepted |
| POP8 | AcadProg | 0.861 | 0.1 | 0.208 | Accepted |
| POP7 | AcadProg | 0.825 | 0.11 | 0.216 | Accepted |
| POP6 | AcadProg | 0.912 | 0.11 | 0.21 | Accepted |
| POP5 | AcadProg | 1.032 | 0.12 | 0.234 | Accepted |
| POP3 | ResearchProg | 1 |  | 0 | Accepted |
| POP2 | ResearchProg | 0.652 | 0.07 | 0.138 | Accepted |
| POP1 | ResearchProg | 0.574 | 0.07 | 0.146 | Accepted |
| LS9R | PartiLearn | 0.681 | 0.17 | 0.342 | Accepted |
| SB13 | Integ | 1 |  | 0 | Accepted |
| SB12 | Integ | 0.791 | 0.14 | 0.274 | Accepted |
| SB11 | Integ | 0.622 | 0.13 | 0.26 | Accepted |
| SB5 | Member | 1.132 | 0.2 | 0.396 | Accepted |
| SB1 | Member | 1.02 | 0.18 | 0.366 | Accepted |
| SB2 | Member | 1.131 | 0.18 | 0.356 | Accepted |

**Table A2.** Discriminant validity proved using chi-square difference in stats tool package.

| Correlation | Overall Model | Chi-Square | Df | *p*-Value | Invariant? | Discriminant Validity |
|---|---|---|---|---|---|---|
| LS<–>SB | Unconstrained | 347.099 | 200 | | | |
| | Fully constrained | 516.305 | 201 | | | |
| | Number of groups | | 2 | | | |
| | Difference | 169.206 | 1 | 0 | NO | |
| SB<–>POP | Unconstrained | 143.148 | 94 | | | Groups are different at the model level. |
| | Fully constrained | 296.914 | 95 | | | |
| | Number of groups | | 2 | | | |
| | Difference | 153.766 | 1 | 0 | NO | |
| LS<–>POP | Unconstrained | 404.494 | 240 | | | |
| | Fully constrained | 542.14 | 241 | | | |
| | Number of groups | | 2 | | | |
| | Difference | 137.646 | 1 | 0 | NO | |

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
