# Peer review of "Development and Validation of Cultural and Academic Experience Questionnaire: A study of East Asian Research Students at Australian Universities"

_education, doi:10.3390/educsci10060148_

Round 1

Reviewer 1 Report

This study presents the validation of an instrument that attempts to measure the cultural and academic perception of international students. 

I feel the overall presentation and methods are adequate. My only concern was the number of individual used in the validation process but I realize it can be difficult to recruit sufficient numbers. 

Author Response

Response to Reviewer 1 Comments

Dear Reviewer,

Thanks for reviewing this manuscript. Your review has really helped us in re-identification and correcting shortcomings in the articles. We have read the article in every detail to improve the English language, grammatical mistakes, errors, omissions and linked all the tables with the texts. Moreover, the short form of the sense of belonging which was “SOB” has been written as “SB” because it saves the reader from linking irrelevant meanings with the sense of belonging.   In-line with your comments, the following changes, have been made. A complete of these changes have been given below.

Point 1: English language and style are fine/minor spell check required 

Response 1: Detailed spell check has been conducted and the following improvements have been made and highlighted.

In the line 7, the word “of” has been removed, hence not highlighted.

In the line 13, words “academic and research progress” added.

In the line 15, word “sector” is added.

In the line 35, words “is needed” are added.

In the line 46-47, “The development of a SB is a significant part of international students becoming socially and academically successful in Australia” has been reworded as “Sense of belonging plays a key role in social and academic success of international students in Australia”.

In the line 70, the word “constituting” is added.

In the line 72, the word “universities” is added.

In the line 84, the addition of few words has been made “Sense of belonging acts as an antidote to the cultural differences because”.

In the line 85-89, improvement has been made as “It is convenient to presume that cultural membership and integration are likely to reduce cultural differences. In the current study, the face and content of SB items indicate the likelihood that the factor analysis of SB will generate the cultural membership and integration as it constructs. In the case of learning strategies”,

In the line 92-93, the improvement made “and may influence learning strategies”.

In the line 141, the word “improved” has been added.

In the line 142, the word “since” has been added.

In the line 211, word “there” has been added.

In the line 212, the word “on” has been added.

In the line 225, the word “that” has been added,

In the line 242, the words “a value” are added.

In the line 244, words “eigenvalues” and “the variance”.

In the line 262, the words “a value” are added.

In the line 264, the word “eigenvalues” has been added.

In the line 265, the words “the variance” are added.

In the line 282, the words “a value” are added.

In the line 285, the word “eigenvalues” is added.

In the line 326, the word “validity” is added.

In the line 337, the words “the correlation” are added.

In the line 375, the word “was” is added.

Kind regards,

Authors

Reviewer 2 Report

The study describes the validation procedure of the CAEQ questionnaire and offers this instrument with special application in higher education settings (PhD students).

Its design is based on an optimal conceptual and instrumental analysis that supports and justifies the explanatory axes of the construct. However, its validation only applies the following techniques: EFA, CFA, convergent validation and discriminant validation. Although supported in the revised scientific literature and other scales close to the concepts under study, an analysis of content validity would have been expected. It is therefore recommended that a brief note on the absence of this type of validation be included.

Author Response

Point 1: An analysis of content validity would have been expected. It is therefore recommended that a brief note on the absence of this type of validation be included.

Response 1: For reference see line 180-198 related to the addition of section "4.2.1. Content and face validation section" has been made in paper as per the guidance of the reviewer.

Reviewer 3 Report

The papper ins interesting. Some recommendatios are:

- Put the objectives and hypotheses of the study in the introduction
- All the tables have to be commented in the text, review table 1.
- Organize the methodology section in the following sections: sample, variables and research design, information collection instruments, data analysis
- The discussion has to go with the results and a conclusions section is missing where the achievement of the objectives and the veracity of the hypothesis are assessed.

Author Response

Response to Reviewer 2 Comments

Dear Reviewer,

Thanks for reviewing this manuscript. Your review has really helped us in re-identification and correcting shortcomings in the articles. We have read the article in every detail to improve the English language, grammatical mistakes, errors, omissions and linked all the tables with the texts. Moreover, the short form of the sense of belonging which was “SOB” has been written as “SB” because it saves the reader from linking irrelevant meanings with the sense of belonging.   In-line with your comments, the following changes have been made. A complete of these changes have been given below.

Point 1: English language and style are fine/minor spell check required 

Response 1: Detailed spell check has been conducted and the following improvements have been made and highlighted.

In the line 7, the word “of” has been removed, hence not highlighted.

In the line 13, the words “academic and research progress” added.

In the line 15, the word “sector” is added.                                                                                                                

In the line 35, words “is needed” are added.

In the line 46-47, “The development of a SB is a significant part of international students becoming socially and academically successful in Australia” has been reworded as “Sense of belonging plays a key role in the social and academic success of international students in Australia”.

In the line 70, the word “constituting” is added.

In the line 72, the word “universities” is added.

In the line 84, the addition of few words has been made “Sense of belonging acts as an antidote to the cultural differences because”.

In the line 85-89, improvement has been made as “It is convenient to presume that cultural membership and integration are likely to reduce cultural differences. In the current study, the face and content of SB items indicate the likelihood that the factor analysis of SB will generate the cultural membership and integration as it constructs. In the case of learning strategies”,

In the line 92-93, an improvement made “and may influence learning strategies”.

In the line 141, the word “improved” has been added.

In the line 142, the word “since” has been added.

In the line 211, word “there” has been added.

In the line 212, the word “on” has been added.

In the line 225, the word “that” has been added,

In the line 242, the words “a value” are added.

In the line 244, words “eigenvalues” and “the variance”.

In the line 262, the words “a value” are added.

In the line 264, the word “eigenvalues” has been added.

In the line 265, the words “the variance” are added.

In the line 282, the words “a value” are added.

In the line 285, the word “eigenvalues” is added.

In the line 326, the word “validity” is added.

In the line 337, the words “the correlation” are added.

In the line 375, the word “was” is added.

Point 2: Put the objectives and hypotheses of the study in the introduction.

Response 2: In 50-63, the research objective has been described as “The research team has explored the vast body of knowledge and conducted an in-depth systematic literature review of the previous scholarship but remained unable to identify any scale which can serve the purpose. All of the searches literature review searches are made from the Google scholar and later from other search engines such as Emerald, JStor, Science Direct, Taylor and Francis, Wiley online library, Blackwell, Ebsco host, Scopus, Elsevier, DOAJ, Copernicus, BASE, Core, Semantic scholar and many other are thoroughly explored. Moreover, Table 2, Table 3 and Table 4 given later in this article, present the detail of the already developed measuring scales in literature. These scales measured the key variables of cultural and academic experiences questionnaires including SB, LS and POP but the research team remained unable to link these scales to the current study. This has led the research team to develop a new scale on the Cultural and academic experiences of East Asian PhD students. Conclusively, t­­­­he objective of this study is to develop a scale to measure the cultural and academic experiences of East Asian PhD students in Australian universities based on their sense of belonging, learning strategies and perception of progress”.

In the line 145-151, the hypothesis of the study has been given as: “An analysis of the previous scholarship indicates that there is a need for a valid and reliable instrument to measure the cultural and academic experiences of East Asian research students at Australian. Following three hypotheses can be drawn from the above debate:

H1: Cultural membership and integration are the valid and reliable constructs of a SOB.

H2: Scientific, artistic, adaptive and participative learning are the valid and reliable constructs of LS.

H3: Academic progress and research progress are the valid and reliable constructs of POP”.

The hypotheses H1, h2 and H3the  have been concluded In the line 360-365 as “Based on the reliability statistics given in Table 2, Table 3 and Table 4 for the constructs of SOB, LS and POP, all of the constructs are reliable. The results of convergent and discriminant validity given Table 9 and Table 10 indicate that all the constructs of SOB, LS and POP hold their respective items converging on their constructs to ensure convergent validity. Similarly, every item in each construct of SOB, LS and POP is different from each other to ensure discriminant validity. Hence, H1, H2 and H3 are proved”.

Point 3: All the tables have to be commented in the text, review table 1.

Response 3: Improvement has been made in linking the Tables to the texts throughout the article. For example:

Table 1 is given in the line 36, 246, 267, 287, 362 and 399.

Table 2 is given in the line 55, 110 and 260.

Table 3 is given in the line 55, 119 and 360.

Table 4 the  in the line 56, 143 and 360.

Table 5 is given in the line 253 and 370.

Table 6 is given in the line 266, 370 and 376.

Table 7 is given in the line 287 and 370.

Table 8 is given in the line 330,339, 350 and 372.

Table 9 is given inthe  line 236, 345, 362, 378,

Table 10 is given in the line 358, 362, 378

Point 4: Organize the methodology section in the following sections: sample, variables and research design, information collection instruments, data analysis

Response 4: Changes have been made. Please see line 153-237.

Point 5:  The discussion has to go with the results and a conclusion section is missing where the achievement of the objectives and the veracity of the hypothesis are assessed.

Response 5: Changes have been made to see line 425-432.

Kind regards,

Authors

Reviewer 4 Report

This is a very interesting study and the results will be of importance to the academic community, particularly as Australia has a high percentage of international students so understanding the importance of helping them to feel they belong and are accepted is critically important. The author/s undertook a large online survey to investigate the perceptions that 211 East Asian research students held of their cultural and academic experiences. The theoretical framework and relevant studies that informed the study are well articulated as were the steps involved in the development of the questionnaire. The exploratory factor analyses identified the items that loaded on each scale and the confirmatory factor analysis confirmed that the measures used in the research were psychometrically sound. The questionnaire that was developed has the potential to provide information to universities that can help them be more responsive to these students needs.

Author Response

Response to Reviewer 3 Comments

Dear Reviewer,

Thanks for reviewing this manuscript. Your review has really helped us in re-identification and correcting shortcomings in the articles. We have article in every detail to improve the English language, grammatical mistakes, errors, omissions and linked all the tables with the texts. Moreover, the short form of the sense of belonging which was “SOB” has been written as “SB” because it saves the reader from linking irrelevant meanings with the sense of belonging.   In-line with your comments, the following changes have been made. A complete of these changes have been given below.

Point 1: English language and style are fine/minor spell check required 

Response 1: Detailed spell check has been conducted and the following improvements have been made and highlighted.

In the line 7, the word “of” has been removed, hence not highlighted.

In the line 13, the words “academic and research progress” added.

In the line 15, the word “sector” is added.

In the line 35, the words “is needed” are added.

In the line 46-47, “The development of a SB is a significant part of international students becoming socially and academically successful in Australia” has been rethe worded as “Sense of belonging plays a key role in the social and academic success of international students in Australia”.

In the line 70, the word “constituting” is added.

In the line 72, the word “universities” is added.

In the line 84, the addition of few words has been made “Sense of belonging acts as an antidote to the cultural differences because”.

In the line 85-89, improvement has been made as “It is convenient to presume that cultural membership and integration are likely to reduce cultural differences. In the current study, the face and content of SB items indicate the likelihood that the factor analysis of SB will generate the cultural membership and integration as it constructs. In the case of learning strategies”,

In the line 92-93, an improvement made “and may influence learning strategies”.

In the line 141, the word “improved” has been added.

In the line 142, the word “since” has been added.

In the line 211, the word “there” has been added.

In the line 212, the word “on” has been added.

In the line 225, the word “that” has been added,

In the line 242, the words “a value” are added.

In the line 244, the words “eigenvalues” and “the variance”.

In the line 262, the words “a value” are added.

In the line 264, the word “eigenvalues” has been added.

In the line 265, the words “the variance” are added.

In the line 282, the words “a value” are added.

In the line 285, the word “eigenvalues” is added.

In the line 326, the word “validity” is added.

In the line 337, the words “the correlation” are added.

In the line 375, the word “was” is added.

Kind regards,

Authors
